# Epigenetic Factors in Eutopic Endometrium in Women with Endometriosis and Infertility

**DOI:** 10.3390/ijms23073804

**Published:** 2022-03-30

**Authors:** Magdalena Adamczyk, Ewa Wender-Ozegowska, Malgorzata Kedzia

**Affiliations:** Department of Reproduction, Poznan University of Medical Sciences, 60-535 Poznan, Poland; ewaoz@post.pl (E.W.-O.); mal.gin@poczta.fm (M.K.)

**Keywords:** DNA methylation, endometrial receptivity, endometriosis, eutopic endometrium, epigenetics, histone modifications, implantation, infertility, TET protein

## Abstract

Eutopic endometrium in patients with endometriosis is characterized by aberrant expression of essential genes during the implantation window. It predisposes to disturbance of endometrial receptivity. The pathomechanism of implantation failures in women with endometriosis remains unclear. This paper aims to summarize the knowledge on epigenetic mechanisms in eutopic endometrium in the group of patients with both endometriosis and infertility. The impaired DNA methylation patterns of gene promoter regions in eutopic tissue was established. The global profile of histone acetylation and methylation and the analysis of selected histone modifications showed significant differences in the endometrium of women with endometriosis. Aberrant expression of the proposed candidate genes may promote an unfavorable embryonic implantation environment of the endometrium due to an immunological dysfunction, inflammatory reaction, and apoptotic response in women with endometriosis. The role of the newly discovered proteins regulating gene expression, i.e., TET proteins, in endometrial pathology is not yet completely known. The cells of the eutopic endometrium in women with endometriosis contain a stable, impaired methylation pattern and a histone code. Medication targeting critical genes responsible for the aberrant gene expression pattern in eutopic endometrium may help treat infertility in women with endometriosis.

## 1. Introduction

The term endometriosis refers to endometrium-like tissue outside the uterine cavity. Clinical and molecular achievements of the last two decades have contributed to the evolution of this mainly anatomical definition. Within the new approach, endometriosis is a complex clinical syndrome characterized by an estrogen-dependent, chronic inflammatory process that affects mainly pelvic tissues [1].

The morphological classification of endometrial lesions includes superficial peritoneal endometriosis (SUP), ovarian endometrioma (OMA) and deeply infiltrating endometriosis (DIE). Pathological changes in the eutopic and ectopic endometrial tissue are responsible for the clinical picture of endometriosis. Endometriotic implants are composed chiefly of stromal cells, and they contain a slight epithelial component that lacks glands that we typically observe in eutopic endometrium.

The majority of women suffering from endometriosis present with typical disease symptoms; however, some of them may be asymptomatic. Endometriosis causes infertility and is the most common cause of chronic pelvic pain in women of reproductive age.

One in ten women at the reproductive age—200 million women worldwide—may have endometriosis [2]. In the group of infertile women, this proportion rises to 20–50% [3]. Endometriosis occurs only in 5% of women without fertility disturbances [4].

According to ACOG (American College of Obstetricians and Gynecologists), the gold standard for diagnosing endometriosis is finding pathological lesions, typical for endometriosis, upon laparoscopy.

The etiopathogenesis of endometriosis remains unclear. It is thought that episodes of retrograde movement of endometrial fragments and blood into the pelvic cavity during menstruation and following implantation and penetration of the peritoneal epithelium are the most reliable endometriosis pathomechanism of the lesser pelvis [5]. Endometriotic implants anchored in the peritoneum obtain a constant blood supply and reveal a suboptimal immunological response essential for their further survival and growth outside the uterine cavity.

The phenomenon of retrograde menstruation is common. Nevertheless, the disease is developed only in some women diagnosed with specific cellular and molecular defects in the tissue of eutopic endometrium or peritoneum [6,7,8]. Such phenomena occur due to specific genetic, epigenetic, environmental and/or immunological factors.

The most probable hypothesis regarding the genesis of endometriosis, proposed by Sampson, states that eutopic endometrium is the critical tissue in the research of this condition [5].

Fertility is defined as a capability to achieve clinically recognized pregnancy [9]. Humans are mammals whose fertility is limited. The average monthly fertility indicator in humans, the probability of pregnancy after one sexual cycle, reaches 15–20% [10]. The embryo’s implantation frequency in the endometrium is estimated to be only 25–30%. Approximately 15% of clinically diagnosed pregnancies are terminated by a spontaneous abortion [11].

Couples that fail to conceive within twelve cycles are considered infertile. The commonly used definition, formulated by the World Health Organization (WHO), defines infertility as a failure to achieve a pregnancy after 12 months of regular intercourse (2–4 times a week), without using any contraceptive methods.

Additionally, 8–12% of couples at the reproductive age face the problem of incapability to achieve pregnancy [12]. Female infertility is responsible for 30% of all causes of a failure to become pregnant. 30% of infertility cases, with an upward trend, are caused by an independent male factor [13]. It is estimated that the standard assessment of fertility cannot reveal abnormalities in 15–30% of infertile couples, which is then referred to as unexplained (idiopathic) infertility [14]. Lastly, in 10% of infertile couples, abnormalities are diagnosed in both partners [14].

Endometriosis is a well-proven cause of infertility. However, the diagnosis of endometriosis may be difficult, particularly in the case of atypical lesions. It is suggested that endometriosis can be undiagnosed and that unexplained infertility can be, in many cases, invisible endometriosis [15,16].

The cause of the difficulty in achieving a pregnancy in women with endometriosis may be elusive and includes immunological, genetic and environmental factors. The pathogenesis of infertility in endometriosis is complex and is likely the sum of abnormalities at many stages of conception. Causes of infertility in all forms of endometriosis may be the following: impaired folliculogenesis, low quality of oocytes, ovulation disturbances, aberrant embryogenesis, or an impaired implantation process.

The fertility indicator in untreated patients with endometriosis is difficult to assess, and according to the data from the literature, it is between 2–10% per one menstrual cycle [17]. The research concerning spontaneous, unassisted fertility in endometriosis patients points to the existence of a correlation between the stage of endometriosis and the chance of becoming pregnant.

In advanced endometriosis, a failure to become pregnant seems to result from the phenotype of the disease [18]. The chronic inflammatory condition in ectopic foci initiates the process of fibrosis and the formation of adhesions, which irreversibly modify anatomical conditions in the lesser pelvis and impair the function of the reproductive organ. The impaired function of fallopian tubes and the inflammatory microenvironment impede the takeover of the released ovum by the fimbriae of the uterine tubes, which negatively influences the conception process and the fallopian transport of the blastocyst to the uterine cavity.

Based on clinical observations and research results, it has been shown that infertility is also a problem in women who suffer from the minimal form of the disease [19]. In mild endometriosis, the leading cause of a failure to achieve a pregnancy seems to be the aberrant endometrium receptivity and, consequently, impaired implantation.

Olive et al. assessed the pregnancy rate in untreated patients with endometriosis depending on the stage of the disease. Half of the patients with minimal and mild endometriosis, 25% with moderate endometriosis, and a small percentage with the severe disease became pregnant within 24 months [20]. Santulli et al. showed a correlation between superficial endometriosis and infertility [21]. The presence of ectopic lesions on the peritoneum threefold reduced the chance of spontaneous fertilization.

The proper progesterone secretion and the response of the endometrium to it during the luteal stage are essential achievements of the proper endometrial receptivity.

Women with endometriosis often present progesterone resistance [22]. The exact cause of this resistance has not been fully explained. The high expression of ERβ and a lower concentration of ERα may be responsible for suppressing progesterone receptors [23]. The diminished expression of progesterone receptors (PR) in the endometrium and the incorrect ratio of the number of A and B progesterone receptors (PR-A:PR-B) may be accountable for progesterone resistance [24,25]. Such a situation favors the increased bioactivity of estrogens in the endometrium due to the lack of progesterone-dependent expression of the enzyme converting estradiol to biologically less potent estrogens (hydroxysteroid 17-beta dehydrogenase 2; HSD17β2) [26]. It has been found that the endometrium of endometriosis patients shows the increased activity or aromatase P450 that metabolizes androgens to estrogens [27]. Research confirms that a failure to obtain the optimal, estrogen-stimulated endometrium in the proliferative stage may lead to abnormal distribution of steroid receptors and contribute to progesterone in endometriosis.

The proper functioning of eutopic endometrium in patients with endometriosis is impaired due to the local domination of estrogen and progesterone resistance. It has been proven that progesterone acts as an anti-inflammatory factor in the endometrium. Progesterone resistance in patients with endometriosis leads to the induction of the pathological inflammatory reaction, the production of E3 prostaglandin, the nuclear transcription factor NF-κB, regulating the immunological response, cyclooxygenase 2 (COX-2), IL-8 and IL-17. It has also been shown that the aforementioned immunological factors secondarily activate the expression of aromatase P450 [28].

Eutopic endometrium in endometriosis patients reveals numerous molecular differences compared to a healthy woman’s endometrium. Research based on the analysis of the results of modern diagnostic methods has found many proteins whose expression in eutopic endometrium in women with endometriosis substantially differs compared to non-endometriosis patients. These substances play a role in cellular proliferation, decidual transformation, angiogenesis, signaling pathways, endometrial receptivity, etc. [26,29].

Dissimilarity in structure and function of endometrium in women with endometriosis predisposes to the disturbances of the receptivity of the uterine mucus and the formation of extrauterine disease. The correlation between implantation failures and the co-occurrence of endometriosis has been proven [30,31].

Thanks to assisted reproductive technology (ART), it is possible to observe the individual reproductive stages in patients with endometriosis: from the maturation of ovarian follicles, through the conception process and the embryonic development, to the process of implantation. The effects of oocyte donation programs have been used to analyze the quality of oocytes and endometrial receptivity in infertility that accompanies endometriosis.

The meta-analysis of the effectiveness of assisted reproductive technology in infertile women with endometriosis has shown that the index of in vitro fertilization (IVF) pregnancies was 50% lower in endometriosis patients than in women with tubal factor infertility [32]. Thus, based on this research, one can assume that the critical factor lowering the IVF effectiveness in this group of patients is the aberrant course of the implantation process. This could be the result of low quality of oocytes, blastemata, and/or endometrial dysfunction. Interestingly, the phenotype of endometriosis (superficial peritoneal endometriosis, endometrioma or deeply infiltrating endometriosis) did not seem to influence the effects of IVF [19]. 

Another study aimed to assess the influence of severe endometriosis on the in vitro fertilization and embryo transfer (IVF-ET) results [31]. Oocytes harvested from healthy donors were used for in vitro fertilization. There were no differences in the scope of the fertilization indicator, the number of transferred embryos, implantation indicator, clinically recognized pregnancies, miscarriages and live births between the groups of infertile patients with stage III or IV endometriosis and healthy individuals.

Leaving aside the tubal factor and providing the optimal environment for in vitro fertilization of the ovum, assisted reproduction technology limits the defective stages of reproduction in patients with advanced endometriosis and seems to increase the chance of successful treatment.

Results show that the potential adverse effect of severe endometriosis on the uterine environment and the implantation process is unclear.

The results of research estimating the effectiveness of IVF in patients with endometriosis are often contradictory and ambiguous.

Previously, it was believed that abnormalities within ova and formed embryos are one of the leading causes of infertility in endometriosis patients. This was suggested by the similar implantation indicator of embryos obtained from the oocyte donation programs in patients with and without endometriosis.

Sung et al. have indirectly pointed to the low quality of oocytes and embryos at all stages of endometriosis [33]. The authors studied the implantation and pregnancy indicators in patients with I-IV stage endometriosis and those without this disease. In patients with minimal, mild, moderate and severe endometriosis, the indicator of clinically recognized pregnancies after the transfer of embryos obtained from oocytes of healthy donors was similar and did not depend on the disease phenotype [33]. The unfavorable influence of endometriosis on the reproductive results, according to the researchers, was not connected to implantation [33].

The latest research review in this field has proven that patients who obtained oocytes from donors with endometriosis reached a low indicator of implantation and pregnancy, while the recipient status did not influence the treatment result [34,35]. Those studies suggest that the diminished reproductive potential of women with endometriosis results from the low quality of oocytes. The detailed analysis of this study allowed a critical interpretation of the presented results. The group of patients participating in the research did not reflect the general population of endometriosis patients. The recipients’ age was higher than the average age of endometriosis patients, which may have resulted in natural, age-related disease limitation. Additionally, participation in the oocyte donation program indicated diminished ovarian reserve and no infertility related to endometriosis. Thus, the presented results of the oocyte donation program do not provide a proper model for assessing implantation and the percentage of pregnancies in young women with endometriosis-related infertility.

Subsequently, the results of other studies point to the relevant disturbances of endometrial receptivity in endometriosis patients as one of the leading causes of infertility in this group of patients.

Prapas et al. designed a study that aimed to assess the influence of endometriosis on implantation, the index of pregnancies and live births in post-menopausal women who underwent the IVF procedure with ova donation. The study results showed that the history of endometriosis in embryo recipients harmed the effectiveness of implantation, even during menopause [30].

Among other mild gynaecological disorders, endometriosis is a disease which causes a diminished potential in reproductively as a result of aberrant endometrial receptivity [36]. This fact confirms observations of the effectiveness of IVF methods. Contemporary techniques of assisted reproduction enable the selection of high-quality embryos. Despite these possibilities, implementation indicators remain relatively low and have not significantly increased during the last decade [37]. Endometrial receptivity is thus essential for successful implantation, and its disturbances reduce the success rate of ART.

The menstrual cycle is regulated by epigenetic mechanisms that maintain the functionality of the endometrium. It has been shown that many genes are involved in the proliferation, differentiation, degeneration and regeneration of uterine mucosal cells [38,39,40]. Due to epigenetic regulation, they undergo increased or decreased expression in the endometrium at individual menstrual cycle stages [29,41,42]. Based on the analysis of gene expression in the eutopic endometrium, it has been revealed that the number of transcriptionally active genes is higher in the secretory phase than in the proliferative stage.

One can assume that infertility in women with endometriosis is, apart from any other known causes, a result of disturbances at the endometrial level. Dysfunction of the endometrium, resulting from its aberrant receptivity, is likely an effect of an impaired pattern of gene expression, that is, the epigenetic pattern, in the eutopic tissue [25].

Since the proper pathomechanism of endometriosis and related infertility remains unexplained, the objective of this paper is to summarise the knowledge on epigenetic mechanisms and epigenetic patterns in eutopic endometrium in the group of patients with both endometriosis and infertility.

The detailed objectives of the study were: analysis the possible relationship between the aberrant pattern of DNA methylation and the histone code to impaired eutopic endometrial receptivity in the group of infertile patients with endometriosis; analysis of the expression of superior epigenetic enzymes in the eutopic endometrium in a group of infertile patients with endometriosis; assessment of the role of ten-eleven translocation (TET) proteins in impaired gene expression in the eutopic endometrium of infertile patients with endometriosis; discussion of the pharmacological strategies to target various epigenetic enzymes in the group of infertile patients with endometriosis.

## 2. Methods

The medical literature was searched using PubMed for the following terms: TET proteins, endometriosis, eutopic endometrium, epigenetic therapy, epigenetics, implantation, DNA methylation, histone modifications, and infertility.

Original papers published in high-citation thematic journals were used, mainly in the last five years. High-quality research projects were selected that used the most modern research methods, representative research material, appropriate size and criteria for the study and control group.

Prospective research on the human eutopic endometrium in vivo, animal model experiments or in vitro studies on eutopic endometrial stromal cell lines were used. The group studied in the analyzed articles were women with endometriosis, especially a subgroup of infertile women. The control group included women without endometriosis. The data obtained for individual phases of the menstrual cycle were mainly used in order to present the data characteristic for the middle secretory phase, i.e., the implantation window.

In particular, the latest papers on the role of TET proteins in the eutopic endometrium and epigenetic therapy strategies in the treatment of epigenetic diseases were analyzed.

## 3. Results

### 3.1. Epigenetics

#### 3.1.1. Epigenetic Mechanisms

Epigenetics is a field of genetics that includes molecular modifications of chromatin that modulate gene expression and genome stability without any changes in the DNA nucleotide sequence. 

Epigenetic modification can “switch on” and “switch off” genes, conditioning their transcription. Epigenetics determines the phenotype of cells. It is one of the key factors that condition cellular differentiation [43].

The epigenetic pattern is transferred to daughter cells during mitotic and meiotic divisions. However, it has been proven that most epigenetic changes occurring in the haploidal karyotype of spermatozoa and ova are removed when two gametes merge. This process is called reprogramming. It allows the foetal cells to create their own epigenetic pattern [44]. At the same time, it has been shown that some epigenetic processes in reproductive cells can omit reprogramming [45]. The transfer of epigenetic information between parents and offspring is called intergenerational epigenetic inheritance. Spontaneously formed human clones, i.e., monozygotic twins, are identical at the level of DNA sequence, but they differ in terms of DNA methylation and the profile of histone modifications. A different epigenetic pattern in a pair of twins with a common genotype influences the individualised susceptibility to diseases and a wide range of anthropomorphic features. It has been discovered that the differences in the content of 5-methylcytosine, changes in its genomic layout, and modifications of the histone acetylation pattern become more profound as the twins’ age progresses [46].

Epigenetics plays an essential role in maintaining the correct, undisturbed development of the organism. Many diseases develop due to changes in the epigenetic pattern occurring at the wrong time or in the wrong place [47].

Epigenetic modifications can be divided into DNA methylation, histone modifications, and microRNA participation. All the epigenetic mechanisms influence one another, they are intertwined, and the observed molecular and clinical effect results from these interactions.

##### DNA Methylation

DNA methylation is the best-known mechanism of epigenetic changes. It is a process of covalent addition of methyl groups to the five positions on the pyrimidine ring of cytosine in the newly sequenced strand of DNA [48]. Methylcytosine (5mC) is the best-characterised epigenetic marker.

Cytosine methylation occurs in dinucleotide sequences of cytosine connected to guanine through the phosphate group; CpG sequences. It is estimated that 70–80% of all genomic CpG sequences are methylated. Recurring CpG dinucleotides form regions of DNA called CpG islands [49]. These regions of the genome have more than 200 base pairs, of which more than half are CpG sequences. In mammals, CpG dinucleotides form 1% of the genome. They also co-create the gene promoter regions of over 60% of all human genes. The majority of them are demethylated. Methylation affects mainly gene promoters of specialised tissues at the initial stages of the development, during cellular differentiation [50]. 

CpG methylation is related to transcriptional silencing of gene expression and chromatin condensation. In this way, it regulates (most often, decreases) the availability of DNA binding sites for various transcriptional factors.

Three DNA methyltransferases catalyse the (deoxyribonucleic acid methyltransferases; DNMT1,2,3) methylation reaction [51]. 

DNMT1 catalyses 97.0–99.9% of the methylation process during mitosis [51]. This enzyme is a component of the replication fork and is responsible for copying the methylation pattern after replication and transferring epigenetic information to daughter cells. DNMT3A and DNMT3B are responsible for DNA methylation *de novo*. They establish a new methylation pattern during embryogenesis. Their role in mature cells has not been fully explained [51].

DNA methylation sites are not random. The methylation pattern is a characteristic feature for a given species, tissue, and even a type of cell. Despite the common belief that DNA methylation pattern is established at the early stages of embryonic development and is maintained throughout life by DNMTs, recent research suggests that active demethylation may be possible in mammalian cells.

DNA demethylation occurs during the reactivation of silenced genes or the correction of improperly methylated bases. Demethylation can be a passive process, dependent on replication when DNMTs does not methylate the newly synthesised DNA strand. As a result, the second replication round, unaccompanied by conservative methylation, yields completely non-methylated DNA [52]. Active demethylation can occur via the enzymatic pathway, irrespective of the course of the cellular cycle. So far, several possible mechanisms of active DNA methylation invertebrates have been proposed. According to the latest research, the pivotal role in the DNA demethylation process is played by TET proteins [53]. Results of the research indicate the possibility of dynamic regulation of DNA methylation. Thus, there is a possibility of reprogramming irreversible processes that define the character of a differentiated cell. 

##### Post-Translational Histone Modifications

Histones play a crucial role in epigenetics. Amine endings of four core histones (H3, H4, H2A, H2B), called histone tails, undergo enzymatic post-translational modifications. They are covalent, usually reversible, adding various functional groups to chosen amino acid residues [53]. The two most common modifications are methylation and acetylation.

In most species, epigenetic modifications affect mainly H3 and H4 histones. Acetylation of H3 histone occurs in lysine (K) 9, 14, 18, 23 and 56; while methylation occurs on arginine (R) 2 and lysine (K) 4, 9, 27, 36 and 79. In the case of the H4 histone, acetylation affects mainly lysine (K) 5, 8, 12, and 16; and methylation—arginine (R) 3 and lysine (K) 20 [54]. Histone tails also undergo modifications to a minor degree: phosphorylation, deamination, ADP-ribosylation, ubiquitination, etc. Thus, there is a vast group of potential modification patterns for a single nucleosome.

Methylation is carried out by histone methyltransferases (HMTs) that bind one or more methyl groups to lysine or arginine residues. It is closely connected to the regulation of transcription. It also affects chromatin architecture, recruitment of transcriptional factors, or interaction with initiation and elongation factors [55]. It plays a significant role in cell differentiation and the organism’s development [56]. For instance, H3K4 methylation plays a part in regulating *HOXA* gene expression during embryogenesis [57].

Scientific data suggest a connection between DNA methylation and histone modifications, particularly lysine (K) methylation [58]. Among others, it has been determined that DNA methylation and H3K9 methylation cooperate in silencing gene expression in the region where they collectively appear [59]. 

The histone acetylation pattern is controlled by maintaining a balance between the activity of histone deacetylases (HDACs) and histone acetyltransferases (HATs). Histone acetyltransferases catalyze the addition of acetyl groups from acetyl-coenzyme A to lysine residues in N-terminal regions of histones to activate transcription. In turn, HDACs are large multiprotein complexes that remove acetyl groups from lysine residues, restoring their positive charge and suppressing transcription. Histone deacetylase inhibitors (HDACIs) manifest an anti-proliferative activity, terminate the cellular cycle and stimulate apoptosis.

Histone modifications condition the availability of DNA for regulator proteins, including the complexes of transcriptional factors in promoter regions. They regulate gene expression, recombination, DNA replication, and the repair and integrity of the genome [54,60]. The local and global organization of chromatin structure is dependent on the change in electrostatic charge supplied with functional groups. The effects of histone modifications may have an antagonistic or synergistic influence on one another. They condition the passage between transcriptionally active and inactive chromatin (euchromatin and heterochromatin, respectively). H3K9 acetylation and H3K4, H3K36 and H3K79 trimethylation cause chromatin activation, while lower acetylation levels and higher levels of H3K9, H3K27 and H4K20 methylation are responsible for the transformation of euchromatin into its inactive form [60]. 

The quantitative assessment of various histone modifications can provide vital information for understanding epigenetic regulations of pathophysiological processes and contribute to the development of pharmacotherapy targeting enzymes that modify histone proteins [61].

##### miRNA Interference

The non-coding transcriptome comprises long non-coding RNA (lncRNA) and microRNA (miRNA) influences general gene expression. LncRNA is a one-strand, non-coding RNA longer than 200 nucleotides. miRNA is a group of small, 20–24 nucleotide RNA molecules that modulate gene expression on the post-transcriptional level. Over 2000 miRNA have been discovered in people, and it is believed that they collectively regulate the level of expression of one-third of genes in the genome [62]. miRNA are connected with many human diseases, and are used for clinical diagnostics and therapeutic targets.

### 3.2. Epigenetics in Endometriosis

Histological assessment of endometrioidal and endometrial tissue has revealed typical features of both. From the molecular point of view, those tissues differ concerning the production of active molecules and the response to steroids, cytokines, adhesive molecules, immunological factors, proteolytic enzymes, their inhibitors, etc. Based on the analysis of the results obtained with the chromatin immunoprecipitation (ChIP) array, it has been shown that endometriotic organoids may become a new, reliable preclinical model for delineating the epigenetic mechanisms underlying endometriosis [63].

The causes of the differences between those tissues are different gene expression, post-transcriptional regulation, and translation products. The level of transcription products of many genes differs from endometriotic stromal cells and the stromal cells in the endometrium in women with and without endometriosis [6]. This discovery prompted scientists to perform epigenomic research in endometriosis. The role of the epigenetic factor in the etiopathogenesis of endometriosis was established based on studies comparing the pattern of gene expression and epigenetic modifications in stromal cells of eutopic and ectopic endometrium.

Endometriotic implants comprise stromal cells whose epigenetic pattern is similar to the phenotype of thecal and granulosa ovarian cells and tissue macrophages [64,65].

Epigenetic modifications in stromal cells, typical for the tissue and somatic mutations in epithelial cells, are found in eutopic and ectopic endometrium. Somatic mutations involve mutations of oncogenes and suppressor genes [66,67]. The change in gene expression pattern in eutopic and ectopic endometrium in patients with endometriosis is stable, suggesting cellular memory participation in its maintenance. Interactions between the epigenetic pattern of stromal cells and mutated genes in epithelial cells remain unchanged.

It has been determined that epigenetic changes in eutopic and ectopic endometrium are responsible for the pathomechanism of endometriosis, and one should assume that they are a cause of related infertility [68,69].

#### 3.2.1. Epigenetics in the Etiopathogenesis of Endometriosis

##### DNA Methylation in Endometriosis

DNA methylation is one of the most common epigenetic modifications in the biology of the endometrium. Numerous studies have revealed a direct correlation between DNA methylation level and the expression of genes connected with implantation [70,71]. We assumed that the correct DNA methylation pattern is responsible for maintaining hormonal balance in the eutopic tissue. Results of studies to date suggest DNA methylation changes in the endometrium at various stages of the menstrual cycle. Ghabreau et al. reported an increased level of DNA methylation during the proliferative stage [72]. Results obtained by Saare et al. indicated differences in endometrial methylome between the mid- and late secretory phase, that is, when the endometrial tissue achieves its maximum thickness, the ability to secrete and readiness for implantation [73].

Aberrant DNA methylation patterns have been found in the endometriotic tissue compared to eutopic endometrium. Distinct methylation of over 40,000 CpG dinucleotides in the genome of endometriotic cells has been confirmed [74]. Scientists have also proven that the expression of one of the DNA methyltransferases (DNMT3B) is visibly changed in stromal cells in endometriosis patients [75]. DNMT3B binds to promoter regions of essential genes in the pathogenesis of endometriosis.

The level of DNA methylation in the whole genome has been compared between eutopic and endometriotic stromal cells [76]. Considerable differences in methylation of 403 genes in both types of cells have been determined. They were mainly genes coding transcriptional factors, *HOXA* genes and genes of nuclear receptors [76]. The specific pattern of DNA methylation/demethylation of stromal cells is responsible for suppressing or overproducing specific proteins. Epigenetic modifications of stromal cells contribute to increased expression and accumulation of inflammatory and tissue-remodeling substances.

A well-studied example of significant epigenetic differences in stromal cells of the endometrium and ectopic foci is the expression of the transcriptional factor—GATA2, GATA6 and steroidogenic factor 1 (SF1).

The change in expression of estrogen receptor-β (ERβ), progesterone receptor (PR), transcriptional factors such as GATA-6 binding factor, SF-1 in endometriotic stromal cells seems to account for estrogen dependency and progesterone resistance, characteristic for endometriosis [77].

GATA2 protein increases the expression of genes involved in the decidual transformation and has a function of progesterone mediator, regulating the metabolism of steroids [78]. *GATA2* gene is hypomethylated in the endometrium and the endometrioidal foci [76]. GATA2 protein is thus abundant in the stromal cells of the endometrium and almost absent in endometrioidal cells. The reverse is valid for the *GATA6* gene. GATA6 and the orphan nuclear receptor SF1 play a crucial role in converting cholesterol to estradiol in endometriotic stromal cells. Their expression is sufficient for transforming stromal cells of the endometrium into endometrioidal cells that excessively synthesize estradiol [28]. SF1 expression is 12,000 times higher in endometriotic stromal cells than in endometrium [79].

##### Histone Modifications in Endometriosis

Recent research shows the significant participation of the histone pattern in modulating gene expression in eutopic and ectopic endometrium tissue in women with endometriosis. Aberrant patterns of histone modifications are likely to play a role in the etiopathogenesis of this disease. So far, a detailed profile of epigenetic lesions of histone proteins in the endometriotic tissue has not been established. Additionally, the role of histone modifications in endometriosis-related infertility remains unknown. 

Results of the global profile of histone acetylation have shown that some histones are hypoacetylated in the endometriotic stromal cells compared to healthy endometrium. This suggests their role in the etiopathogenesis of this condition [80].

HDAC enzymes deacetylate residues of crucial amino acids in histone tails, changing chromatin conformation and suppressing transcription. HDACs act as transcription repressors. Out of 18 identified HDACs, two of them—HDAC1 and HDAC2—belong to the Sin3a repressor complex. HDAC1 and HDAC2 are conservative enzymes that regulate the course of the cellular cycle. Their expression correlates with the nuclear protein level related to Ki67 cellular proliferation [81].

Munro et al. have proven that the stage of histone acetylation increases during intensified transcriptional activity in the endometrium. It indicates the hormonal regulation of the acetylation process and histone deacetylation in the eutopic tissue [82]. During the whole menstrual cycle, HDAC1 expression in the endometrium remains constant. The level of HDAC2 transcript is slightly elevated in the secretory phase [83].

Colon-Diaz et al. assessed the fundamental and hormonally regulated expression of HDAC1 and HDAC2 and the level of their protein products in the line of stromal cells of the eutopic endometriotic tissue [84]. The essential expression of both HDAC was higher in the ectopic tissue than in the endometrium in non-endometriosis patients. The authors have demonstrated that endometriotic cells lost the hormonal modulation of *HDAC1/2* gene expression. HDAC1/2 expression level has also been compared in frozen biopsy specimens of fresh tissues. The highest expression of both *HDAC* genes was observed in the endometrium of endometriosis patients.

A reduced expression of the HDAC3 enzyme has been demonstrated in the eutopic endometrium of infertile women with endometriosis compared to the control group. Loss of HDAC3 activity in the mouse model resulted in a decidualization defect and infertility due to implantation failure. Incorrect transcriptional activity of target genes for HDAC3 such as *COL1A1*, *COL1A2* contributed to the overexpression of collagen type I in the endometrium. The hypothesis was that *HDAC3* is a critical gene for endometrial receptivity in eutopic tissue of infertile women with endometriosis [85].

HDAC enzyme plays a dominant role in the epigenetic regulation in women with endometriosis [86]. Research results enable us to think that the etiology of endometriosis can be explained by the epigenetic deregulation of gene expression resulting from the disturbance of HDACs expression.

Differences have been found in the histone patterns between stromal cells of the endometrial, ovarian cyst and the endometrium of women who do not have endometriosis. The H3 and H4 histone acetylation levels were lower in the ectopic tissue than in the eutopic tissue of healthy women [80]. Xiaomeng et al. have assessed the global H3 and H4 histone acetylation in endometriotic lesions, eutopic endometrium in patients with and without endometriosis. They have found that H4 histone hypoacetylation occurs in ectopic tissues and eutopic endometrium in women with endometriosis compared to a healthy woman’s endometrium [87]. In turn, Monteiro et al., studying the acetylation of histone proteins in the endometrium of women with and without endometriosis, have shown H3 histone hypoacetylation in the endometrium of women with endometriosis [88].

During their assessment of histone methylation, Xiaomeng et al. have observed global hypomethylation of active chromatin markers, i.e., H3K4 and H3K9, in endometriotic lesions and eutopic endometrium in endometriosis patients [87]. Monteiro et al. have presented results contradictory to those obtained by Xiaomeng, showing that H3K4, H3K9 and H3K27 were hypermethylated in eutopic endometrium in endometriosis patients in comparison with the group of healthy women. Thus, the histone methylation profile in the tissues of eutopic and ectopic endometrium in endometriosis patients’ needs further research.

##### miRNA Interference in Endometriosis

In endometriosis patients, constant patterns of miRNA expressions have been observed compared to in non-endometriosis patients. A different the expression of over 50 miRNA has been found in the endometriotic cells [89]. However, data concerning their implantation role remain ambiguous. The data show that the differentially expressed circRNAs might be potentially involved in pathophysiology of endometriosis-associated infertility [90].

#### 3.2.2. Epigenetics in Endometriosis-Related Infertility

The analysis of the molecular background of unsuccessful implantation in women with endometriosis is based on comparing the expression of chosen genes during the implantation window (or at any other stage of the menstrual cycle) in women with and without endometriosis.

The gene expression level during the implantation window was analyzed in the endometrium of women with and without endometriosis [91]. Ninety-one genes were identified whose expression was increased, and there were 115 genes whose expression was diminished in eutopic endometrium in women with endometriosis compared with women without this condition. The indicated genes were compared with others whose expression is hormonally regulated in healthy women’s endometrium [29]. Based on this analysis, 12 target genes were identified and divided into three groups [91]. The first group included eight genes, the expression of which under physiological conditions was increased during the implantation window and decreased in women with endometriosis. Three genes in the second group showed a diminished level of transcription during the implantation window in healthy women and an increased level in the endometrium of women with endometriosis. The third group was comprised of one gene whose expression was diminished during the implantation window under physiological conditions and even more diminished expression in endometriosis patients.

Candidate genes were indicated, and they promote the endometrial environment that is unfavorable for implantation. They include, among others, apoptosis genes, ion transporters, immunological modulators, secretory proteins, membrane proteins, transcriptional factors, and others. Their aberrant expression reduces the chance of successful implantation due to an immunological dysfunction, inflammatory reaction and apoptotic response. 

Women with endometriosis were found to have the disturbed expression of many implantation markers in the eutopic endometrium, which may lower their reproductive potential. The trial confirmed a low expression of the leukaemia inhibitory factor (LIF), interleukin 11 (IL-11R), and a decreased level of the enzyme involved in the synthesis of endometrial ligand for L-selectin in eutopic endometrium in the group of infertile women with endometriosis, compared with the fertile control group [92,93,94,95].

The prospective, double-blinded trial has shown that patients with stage one or two endometriosis present with a diminished expression of the cellular adhesive molecule—integrin αvβ3—during the implantation window [96]. The pregnancy rate in infertile women with endometriosis undergoing in vitro fertilization was compared depending on the expression level of integrin αvβ3 in the eutopic endometrium [97]. IVF was less effective in the group of patients with low expression of integrin αvβ3. Further analysis identified 400 genes and more than 1300 genomic regions that differ in the level of mid-secretory methylation in the group of women with low expression of integrin αvβ3 compared to the control group. The overlap of gene and methylation array data identified 14 epigenetically dysregulated genes. Additionally, hypomethylation and over-expression of the gene for the aryl hydrocarbon receptor (AHR) protein were detected in this group of women. The increase in AHR expression and altered methylation of 14 indicated genes, according to the authors, may be a diagnostic tool to identify a subgroup of women with endometriosis-related infertility.

Another study assessing gene expression in eutopic endometrium through tissue microarrays has shown the aberrant amounts of protein products of target genes for progesterone in endometriosis patients [98]. *HOXA* genes are one example of such genes. The diminished expression of *HOXA* genes causes a diminished expression of dependent genes that code the mediators of endometrial receptivity, such as integrin αvβ3 or IGFBP-1. Under normal conditions, high expression of the *HOXA10* gene suppresses the transcription of the human homologue gene of empty spiracles in *Drosophila melanogaster* (the Empty Spiracles Homeobox 2; Emx2/EMX2), which is connected to aberrant implantation. In connection with diminished expression of HOXA10 in the endometriosis, the increased level of endometrial mRNA EMX2 is found during the peri-implantation period, which may result in the aberrant course of the embryo’s implantation process [99]. In the promoter region of the *HOXA10* gene, specific epigenetic modifications in the secretory phase in patients with endometriosis have been identified [100]. The suppression of the *HOXA10* gene, which naturally modulates the action of progesterone, may additionally favor progesterone resistance and contribute to impaired implantation in women with endometriosis.

The latest research shows that molecular diagnostics can explain the increased pregnancy rate in infertile women with endometriosis treated with letrozole. In an animal model, letrozole was shown to increase the expression of integrin αvβ3 and HOXA10 in the eutopic endometrium and endometrial receptivity [101].

Tests on humans and animals have shown that *HOXA10* hypermethylation is one of the most important mechanisms responsible for its diminished expression [25,102]. In patients with endometriosis, *HOXA10* gene hypermethylation was confirmed in three regions, rich in CpG islands in eutopic endometrium [99]. Excessive *HOXA10* methylation may also result from the increased expression of the gene coding DNA 3A methyltransferase (*DNMT3A*) [103].

It has been proven that the stromal cells of eutopic endometrium in women with endometriosis are also characterized by high levels of transcripts and protein products, among others, for aromatase, cyclooxygenase (COX) 2 and interleukin 6 (IL-6) [64,104,105]. 

The expression of genes connected with women’s fertility with endometriosis is altered due to the change in the methylation pattern of their promoter regions.

The latest research suggests that the aberrant gene expression in stromal cells of eutopic endometrium is an effect of epigenetically incorrectly programmed and aberrantly differentiating mesenchymal endometrial matrix cells [106]. Currently, it is likely that the explanation of the role of epigenetic mechanisms, the role of the endometrial matrix cells and their interaction is key to the pathology of endometriosis and related infertility.

### 3.3. The Role of TET Proteins in Endometriosis

#### 3.3.1. The Function of TET Proteins in Regulating Gene Expression

TET proteins are enzymatic proteins that function as dioxygenases. Three TET (1,2,3) proteins have been identified in human cells coded by three distinct genes [107]. They influence epigenetic modifications depending on their enzymatic activity and histone protein modifications. They play a chief role in regulating gene expression and cellular differentiation.

In the embryonic cells of mice, a double role of TET1 protein has been determined regarding the regulation of gene transcription: activating and repressing.

The most significant function of TET proteins, resulting from their enzymatic activity, is their participation in active DNA demethylation. The family of TET enzymes catalyzes the transformation of 5-methylcytosine (5mC) to cytosine through three subsequent oxidation reactions. TET1 protein is essential for maintaining the hypomethylation state of promoters of transcriptionally active genes. However, it also participates in suppressing transcription via direct binding of protein complexes to chromatin and histone modification [108].

TET1 protein participates in suppressing some genes’ expression by recruiting the Polycomb repressive complex 2 (PRC2) to promoters, rich in CpG pairs [108]. Moreover, TET1 protein directly binds with the Sin 3 transcription regulator family member A, which, using forming a complex with HDAC, accounts for the repression of transcription [109]. 

The expression of *TET* genes is regulated transcriptionally, post-transcriptionally and post-translationally. Relatively little is known about the control of *TET* genes’ transcription. Recent studies have shown a relationship between decreased levels of miRNA22-5p and increased expression of the *TET2* gene in the endometrium during the implantation window in patients with endometriosis [110]. All the regions of the *TET* promoter include CpG islands that can be potentially suppressed by DNA methylation.

#### 3.3.2. The Function of TET Proteins in Eutopic Endometrium

The role of TET proteins in the pathology of the endometrium has not been determined in detail. So far, a few studies have been published that assess the level of TET gene expression in the uterine mucous membrane [111]. One study included infertile endometriosis patients [112]. Ciesielski et al. assessed the expression of TET proteins in endometrial cancer [113].

Mahajan et al. have presented the results of the assessment of *TET1* gene expression in eutopic endometrium in the population of healthy women. They have shown a higher level of TET1 transcript in the mid-secretory phase than in the proliferative phase. A high expression of gene *TET1* in the secretory phase in healthy women correlates with the high expression of genes involved in obtaining the optimal endometrial receptivity. The scientists identified the main compartment of the intensified, estrogen-dependent *TET1* gene expression in the eutopic tissue, comprised of stromal cells of the endometrium [114]. It is likely that DNA hydroxymethylation, mediated by TET protein, could be a factor influencing hormonal instability seen in endometriosis [115].

Roca et al. were the first who compared *TET* gene expression in eutopic and ectopic endometrium in patients with and without endometriosis [111]. Endometriotic foci were determined to have the lowest expression of tested genes. The authors obtained a low level of mRNA of *TET* genes in the eutopic tissue of endometriosis patients compared to non-endometriosis patients. However, this difference was not statistically significant. 

Another subject of those authors’ study was the identification of the main cellular compartment responsible for *TET* gene expression in the endometrium. They assessed the level of TET transcript *in vitro* in the culture of fibroblasts of endometrial stromal cells. Similarly to the studies *in vivo*, the mRNA level of *TET* genes was lowered in the group of endometriosis patients versus the group without this condition. In this case, differences were statistically relevant, which may have resulted from the selection of endometrial cells that were to the greatest extent responsible for the changes in *TET* gene expression in the tissue. The stromal expression of *TET* genes seems thus to be responsible for the aberrant expression of *TET* genes in eutopic endometrium.

In order to verify the hypothesis mentioned above, Roca et al. studied *TET* gene expression in the cell line of the endometrial epithelium. The expression of *TET* genes in epithelial cells turned out to be 10–30 times higher than the one in stromal cells, thus confirming the authors’ assumption.

Szczepańska et al. assessed *TET* gene expression and the level of TET proteins in eutopic endometrium in infertile endometriosis patients and fertile, healthy women [107]. They observed a significantly diminished level of TET1 transcripts and TET1 protein in infertile endometriosis patients compared with fertile women without this condition. The authors revealed differences between studied groups concerning the mid-secretory phase. During the proliferative phase, the transcript and *TET1* gene protein product did not differ between women with and without endometriosis. Additionally, the division of endometriosis patients according to the stage of the disease, based on r-ASRM classification, did not influence the obtained results. TET1 expression in infertile women with a mild and advanced form of the disease was lower than in the fertile control group. 

The objective of Ciesielski et al.’s study was to assess the expression of TET1, TET2 and TET3 in endometrial cancer, concerning the clinical and pathological features of the neoplasm [113]. The analysis showed that the average mRNA level of TET1 and TET2 was considerably lower in the neoplastic tissue than in the samples of the unchanged endometrial tissue. According to FIGO, TET1 and TET2 expression were lower than in neoplasms classified as stage I and II in more advanced tumors. The authors observed low TET1 and TET2 in tumors capable of metastasizing to regional lymph nodes compared to patients with non-dissipated neoplastic disease. The survival period of women with a low level of TET1 transcript was shorter than that of patients with a high expression level. Based on obtained results, the authors have proven that the level of TET1 gene expression is an independent prognostic factor of survival in endometrial cancer.

## 4. Epigenetic Treatment in Endometriosis 

Epigenetic mechanisms are inherited, dynamic, but reversible modulators of gene expression. Reversibility is a very important feature of epigenetic modifications as it allows for targeted pharmacological treatments, or “epigenetic therapies”. The goal of treatment is to reverse the abnormal epigenetic patterns in the affected cells. This is possible by altering the activity of the enzymes involved in establishing the methylome and the histone pattern in the pathological tissue. The target enzymes of epigenetic drugs include DNMTs, HDACs, histone acetyltransferases, histone methyltransferases and histone demethylases [116].

It has been shown that deregulation of the expression of the following epigenetic enzymes may play a key role in the etiopathogenesis of endometriosis and related infertility: overexpression of DNMT3B, *HDAC1*, *HDAC2* and decreased expression of *HDAC3* and *TET1*.

To date, the best known potential epigenetic drugs in endometriosis are nucleoside and non-nucleoside DNMTs inhibitors. During replication, these nucleoside inhibitors integrate into the nucleotide chain replacing cytosine and block the action of DNA methyltransferases in the S phase of the cell cycle. They cause DNA hypomethylation and reactivation of the expression of previously silenced genes [117,118]. Treatment with a demethylating agent (DMA) has been shown to significantly increase ERβ mRNA and aromatase levels in endometrial cells and may indicate a possible epigenetic therapeutic target [23,118]. Importantly, administration of demethylating drugs causes a long-lasting demethylation effect, even after a short period of treatment.

However, nucleoside inhibitors of DNMTs such as 5-azaC and 5-aza-dC are characterized by a lack of selectivity for individual DNMTs isoforms (i.e., DNMT1 vs. DNMT3s), leading to an increase in side effects. Therefore, the drug targeted in endometriosis will be a specific inhibitor of the DNMT3B isoenzyme. This substance, apart from selectivity, should be characterized by high bioavailability, stability also in aqueous solutions of acids and bases, and should not be a substrate of other enzymes [119].

Histone deacetylase inhibitors (HDACI) are a promising group of drugs in the treatment of endometriosis [80]. HDACIs in the stromal cells of endometriotic cysts have been shown to reactivate or silence critical genes for the pathogenesis of endometriosis. The change in the degree of acetylation of regulatory regions resulted in inhibition of cell proliferation, cell cycle arrest and induction of apoptosis in ectopic tissue cells.

HDAC1, HDCA2 are two of the four enzymes in class I HDACs. They contain a zinc-dependent catalytic center. Selective inhibitors of HDAC1 and HDAC2 isoenzymes have zinc binding domains, which guarantees high binding affinity of the inhibitor to the HDAC target site [120]. An example of such a substance is hydroxamic acid. Its synthetic derivatives can increase their bioavailability due to more rigid sulfonamide linkers. Panobinostat is the only one EU-approved HDAC inhibitor to date [121]. More than 10 other HDACs inhibitors of hydroxamic acid are currently in clinical trials [122]. However, hydroxamic acids can be metabolized to toxic products and may be off-target due to non-selective metal binding. Promising results can be expected after therapy with *ortho*-aminoanilides (benzamides). Benzamides show selectivity against HDAC class I. Chinamid, a benzamide derivative, has been shown to be effective in inhibiting HDACs 1,2,3, and the class II HDAC 10 at a nanomolar level [123].

The decreased level of HDAC3 in the eutopic endometrium of infertile patients with endometriosis forces the search for a substance that selectively inhibits HDAC1/2 and activates the HDAC3 enzyme.

The combination of HDACs inhibitors and DNMTs inhibitors has been shown to be beneficial in patients with solid tumors [124,125]. Currently, the combination of epidrugs and immunotherapy also shows promising results in the treatment of cancer patients [126,127].

Moreover, it would also be interesting to determine whether the manipulation of the activity of TET proteins by modulators or small molecules could be of use in the treatment of endometriosis-related infertility. Activation of the *TET1* gene may be a specific and innovative goal of epigenetic therapy in endometriosis. Our unpublished results show that hypermethylation of its promoter region is an effective inhibitor of *TET1* gene expression in the eutopic endometrium in women with endometriosis. In this case, also DNMTs inhibitors can contribute to the activation of the silenced *TET1* gene. Future research will aim to determine which type of DNMTs isoenzyme is responsible for the hypermethylation of the *TET1* promoter region. Moreover, it may be possible in the future to develop synthetic enzymes that demethylate short sections of the genome, such as the regulatory region of the TET1 gene.

The activity of TET proteins has also been shown to be regulated by small molecules. Ascorbic acid increases the activity of the TET1 protein by reducing Fe (III) to Fe (II) at its catalytic center [128]. Additionally, it has been shown that L-cysteine and hydroxychloroquine, a compound with strong reducing properties, increase the level of 5-hmC in the eutopic endometrium by increasing the activity of the TET1 protein [128,129]. Ascorbic acid, L-cysteine and hydroxychloroquine analogs could potentially increase the activity of the TET1 protein.

Based on the available evidence, endometriosis as an epigenetic disease can be treated with drugs that alter gene expression. Epigenetic drugs are in clinical trials. Their long-term safety and side effects still need to be confirmed before using these drugs in everyday practice.

## 5. Summary

The review presents the latest achievements in molecular research related to the epigenetic basis of endometriosis-related infertility. The special role of superior regulators of gene expression in eutopic tissue in the group of infertile patients with endometriosis was summarized. In addition to DNA methyltransferases and histone modifying enzymes, the special role of the recently discovered multifunctional TET proteins was highlighted. TET enzymes regulate gene expression in the middle secretory eutopic endometrium, which may be of key importance in the etiopathogenesis and, in the long term, in the treatment of endometriosis-related infertility. So far, individual papers on the role of TET proteins in this group of women have been published.

The molecular achievements of the last decade have been used to analyze therapeutic options in endometriosis and related infertility. The latest literature reports on the current possibilities of epigenetic therapy in other fields of medicine, especially in oncology, were analyzed. The pharmacological strategies to target various epigenetic enzymes implicated in endometriosis have been proposed. Particular attention was paid here to the TET1 protein, which could become the target of epigenetic therapy in future research protocols in the group of infertile women with endometriosis.

The genetic–epigenetic theory of the etiopathogenesis of endometriosis combines all theories proposed to date. It suggests that molecular abnormalities result from hereditary genetic and epigenetic changes [130]. Scanning the whole genome and the analysis of interconnections have allowed genes to be identified that are potentially involved in its development.

Bouquet De Joliniere et al. and Signorile et al. suggested the embryological background of endometriosis [131,132]. They propose that failures in establishing proper patterns of DNA during the critical embryonic development period might cause endometriosis.

Endometrial cells, containing a stable, aberrant methylation pattern and histone code, implant outside the uterine cavity, forming microscopic lesions on the surface of the peritoneum and ovaries. Endometriotic lesions rely on hormonal, immunological and growth factors in the peritoneal cavity and plasma. Additional, acquired genetic and epigenetic modifications are necessary for developing ovarian and severe endometriosis. Recurrent episodes of tissue damage during micro bleeding incidents, inflammatory conditions and oxidative stress may cause further changes in ectopic foci and contribute to developing the advanced form of endometriosis [133,134].

Changes in eutopic endometrium are common for all stages of that disease, and they equally limit fertility and implantation. The role of epigenetics, endometrial matrix cells, and their interactions is crucial in understanding the mechanisms of this disease and related infertility. 

Endometriosis is an epigenetic disease, meaning that aberrant patterns of gene expression are responsible for the clinical consequences of endometriosis. Macroscopically unchanged eutopic tissue, as a result of pathological silencing or activation of crucial genes for implantation, loses its most important function—providing conditions for blastocyst implantation. Superior epigenetic enzymes, including DNMT3B, HDAC1/2/3, and TET1, may be responsible for the aberrant DNA methylation pattern and the histone code in the eutopic endometrium. Treatment of endometriosis is insufficient. It is not effective in preventing relapse or worsening of the disease. In vitro fertilization methods give a chance of having children in infertile couples. The causal treatment of endometriosis, such as epigenetic treatment, could be a chance to reverse the aberrant epigenetic changes in the endometrium in women for whom the current treatment has not brought the expected benefits (Figure 1).

## 6. Conclusions

In endometriosis, we established aberrant expression of many crucial genes during the implantation window in eutopic endometrium, which is likely to result from impaired epigenetic mechanisms in their promoter regions. 

In endometriosis, impaired methylation and histone code in eutopic endometrium may promote the endometrial environment that favors implantation due to an immunological dysfunction, inflammatory reaction, and apoptotic response.

Medication targeting crucial genes responsible for the aberrant pattern of its expression in eutopic endometrium may help treat infertility in women with this disease.

## Figures and Tables

**Figure 1 ijms-23-03804-f001:**
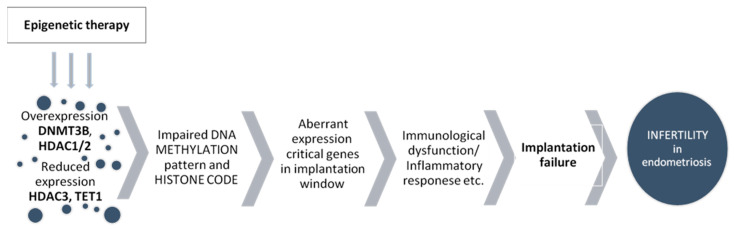
The epigenetic background of implantation failures in the pathogenesis of endometriosis-related infertility. Targets of epigenetic therapy in infertile women with endometriosis. DNMT—DNA methyltransferase, HDAC—histone deacetylase, TET1—ten-eleven-translocation 1 protein.

## Data Availability

Not applicable.

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
