# Peer review of "Epigenetic Factors in Eutopic Endometrium in Women with Endometriosis and Infertility"

_ijms, 2022, doi:10.3390/ijms23073804_

Round 1

Reviewer 1 Report

In the present review the authors aimed at summarising the molecular mechanisms underlying epigenetic modificcations occuring in the eutopic endometrium and affecting embryo implantation capacity in women with endometirosis. The manuscript is clear and well written and represent the most comprehensive review of the topic present in literature and is worth of publication in the current form.

Author Response

Thank you very much for the submission review. We are very pleased that the manuscript has been positively assessed. 

Reviewer 2 Report

It is a very valuable manuscript that gives knowledge about the molecular pathomechanisms of endometriosis. I found only some minor issues, which are rested below:

Please do not use the subsection in the introduction.

The objective should also be included in the introduction section

In objectives, please describe more clearly the main aims of the paper.

To summarize it will be great to prepare some table o graphic scheme.

Reviewer 3 Report

[review text omitted: it was posted to a different submission]

Reviewer 4 Report

The authors have discussed in detail about the various epigenetic mechanisms in eutopic endometrium in the group of patients with both endometriosis and infertility. Additional suggestions for improvements are as follows:

  1. The novelty of the article should be clearly highlighted as number of excellent reviews have already been published on this topic.
  2. More references from last few years should be added to improve visibility and quality of current work.
  3. The search strategy used for the literature review should be indicated.
  4. The pharmacological strategies to target various epigenetic enzymes implicated in endometriosis should be discussed.
  5. The authors should provide their own justification and relevance of the study. This will help the readers to understand the importance of the paper.
  6. The manuscript should be carefully checked for typographical errors.

Author Response

This manuscript is a resubmission of an earlier submission. The following is a list of the peer review reports and author responses from that submission.

Round 1

Reviewer 1 Report

The review summarizes epigenetic mechanisms in eutopic endometrium. I believe this topic is not interesting and difficult to follow  for the readers of IJMS. The topic falls into medicinal category and should be submitted to a more specialized journal in that field. Also, English should be extensively revised since many partes are difficult to understand

Author Response

Thank you for reading our manuscript and valuable comments on it.

            The topic of our study is the molecular basis of two diseases that are probably linked by epigenetic pathogenesis. Infertility is a civilization disease of the 21st century. Endometriosis is a disease that affects even every 10th woman in the world. The actual incidence of endometriosis is unknown. This is due, among other things, to the lack of a non-invasive and sensitive diagnostic method. Laparoscopy is still the gold standard in the diagnosis of endometriosis.

            Endometriosis is an enigmatic disease that raises many questions. We still do not know an effective treatment for this disease. The epigenetic background of endometriosis is currently the subject of intense research. The results of molecular studies of the epigenetic code in the eutopic endometrium indicate significant differences in the methylome and the histone code between the group suffering from endometriosis and the group without endometriosis.

            The International Journal of Molecular Sciences is a journal dedicated to studies in the field of molecular biology, cell biology, molecular medicine. The Epigenetic Society is also associated with IJMC.

            Epigenetics research is currently a particularly interesting research topic in many scientific disciplines, especially in medicine. We hope that the issue we have developed will be an interesting and inspiring topic for the readers of the journal.

            To meet the possible difficulties associated with professional terminology in the field of epigenetics, the first part of the review was devoted to discussing the basic issues in this subject. We presented in the most accessible way the most important information on epigenetic mechanisms: DNA methylation, histone modification, miRNA interference, and discussed the most important enzyme proteins involved in chromatin modifications that lead to changes in gene expression. We believe that the proper names of genes and proteins specific for endometrial receptivity and endometriosis will be known to researchers interested in the discussed issue.

            The work was again subjected to linguistic analysis. The working language has been simplified. We believe that these changes will contribute to a clearer message of the issues discussed. All introduced modifications can be tracked with the use of the 'track changes' tool.

            Thank you again for your comments on the manuscript. We hope that the introduced changes will improve the quality of the work, increase its readability and become a source of inspiration for further scientific discoveries in this field of medicine.

Reviewer 2 Report

Epigenetic factors in eutopic endometrium in infertile women with endometriosis may be very important to introduce newe treatment modalities in the future. Association with decreased endometrial αvβ3 integrin expression for endometrial receptivity  should be also analyzed. Key epigenetic changes should be highlighted from clinical point of view.

Round 2

Reviewer 1 Report

I appreciate the effort made by the authors. However, I have not changed my mind. As indicated in my first report, I believe the topic falls into medicinal category and should be submitted to a more specialized journal in that field.

Reviewer 2 Report

x